# Green Synthesis and Characterization of Silver Nanoparticles with High Antibacterial Activity Using Cell Extracts of Cyanobacterium *Pseudanabaena/Limnothrix* sp.

**DOI:** 10.3390/nano12132296

**Published:** 2022-07-04

**Authors:** Dimitra Karageorgou, Panagiota Zygouri, Theofylaktos Tsakiridis, Mohamed Amen Hammami, Nikolaos Chalmpes, Mohammed Subrati, Ioannis Sainis, Konstantinos Spyrou, Petros Katapodis, Dimitrios Gournis, Haralambos Stamatis

**Affiliations:** 1Department of Biological Applications and Technology, University of Ioannina, 45110 Ioannina, Greece; karagdimitra@hotmail.com (D.K.); theo.tsaki10@gmail.com (T.T.); hstamati@uoi.gr (H.S.); 2Department of Materials Science and Engineering, University of Ioannina, 45110 Ioannina, Greece; pzygouri@gmail.com (P.Z.); chalmpesnikos@gmail.com (N.C.); mossubrati@gmail.com (M.S.); 3Department of Materials Science and Engineering, Cornell University, Ithaca, NY 14853, USA; mah424@cornell.edu; 4Cancer Biobank Center, University of Ioannina, 45110 Ioannina, Greece; isainis@uoi.gr

**Keywords:** cyanobacterium, *Pseudanabaena*/*Limnothrix* sp., biosynthesis, silver nanoparticles, antibacterial activity

## Abstract

In this work, we demonstrated the ability of the cyanobacterium *Pseudanabaena/Limnothrix* sp. to produce ultra-small silver nanoparticlesin the forms of metallic silver (Ag^0^) and silver oxides (Ag_x_O_y_) via a facile green synthetic process. The biological compounds in the cyanobacterial cellular extract acted both as reducing agents for silver ions and functional stabilizing agents for the silver nanoparticles. Furthermore, the antibacterical activity of the as-synthesized nanoparticles against Gram-negative *Escherichia coli* and Gram-positive *Corynebacterium glutamicum* bacterial cells was evaluated. The experimental results revealed a remarkable bactericidal activity of the nanoparticles that was both time-dependent and dose-dependent. In addition to their excellent bactericidal properties, the developed nanoparticles can be used as nanosupports in various environmental, biological, and medical applications.

## 1. Introduction

Metallic and metal oxide nanoparticles have gained significant attention as they exhibit enhanced properties, compared to their bulk forms [1]. Specifically, silver nanoparticles (AgNPs) have emerged as outstanding antimicrobial nanomaterials exhibiting strong bactericidal activity against many species of bacteria, including Gram-positive and Gram-negative strains. The bactericidal effect of silver ions (Ag^+^) and AgNPs has been extensively researched and tested on different bacterial strains, such as *Staphylococcus aureus*, and *Escherichia coli* (*E. coli*) [2]. Moreover, it is worth mentioning that biogenic AgNPs are more effective than classical antibiotics as microbes develop much lower resistance to them [3].

Even though the exact mechanism is not known, Ag^+^ ions are capable of penetrating the bacterial cellular membrane, thereby triggering a number of bactericidal processes, such as oxidative stress, enzyme deactivation, and interference with DNA replication, hence ultimately leading to apoptosis (i.e., programmed cell death) [4]. Moreover, owing to the high surface-to-volume ratio of AgNPs, which significantly exceeds that of bulk silver, one would intuitively anticipate the former to be more effective bactericidal agents.

There are many synthetic routes for the preparation of AgNPs (in the forms of AgO, AgO, and Ag_2_O). The most common include chemical reactions [5], where the nanoparticles are synthesized in solution, physical methods [6], and photochemical synthesis [7], all of which are costly, energy-consuming, and environmentally unfriendly. To overcome the environmental barrier, significant efforts have been devoted recently toward green synthesis of AgNPs [8] by utilizing natural reducing agents present in cellular extracts of microorganisms. These natural reducing agents also can act as colloidal stabilizing agents [9]. In this way, the microorganisms serve as bioreactors, thus ensuring an environmentally friendly approach for the preparation of stable AgNPs [10]. In this regard, cyanobacteria constitute a wide range of microorganisms capable of growing in many different environments [11]. An interesting category of cyanobacteria are filamentous cyanobacteria, which are characterized by their ability to reduce the organic load of their environment for their growth. This could be utilized in cleaning burdened environments while simultaneously producing high-value products (nanomaterials). The *Pseudanabaena/Limnothrix* sp. strain, named Pamv7, is a filamentous strain that has been isolated from Lake Pamvotis in Northwestern Greece, an area which suffers from eutrophication [12].

Biosynthesis of AgNPs follows two general routes: (i) exogenous (extracelluar synthesis) and (ii) endogenous (intracellular synthesis). Specific accessory pigments from cyanobacteria seem to play a crucial role in the synthesis of AgNPs. For example, phycocyanin, which is a blue-colored pigment belonging to the phycobiliprotein family, is able to reduce Ag^+^ ions to form AgNPs [13]. In the present work, we report an eco-friendly synthesis of AgNPs using the cellular extract of the *Pseudanabaena/Limnothrix* sp. cyanobaterial strain. The produced AgNPs are characterized by FTIR, XPS, AFM and TEM to examine their chemical, structural, and morphological properties. The bactericidal effect of these nanoparticles on Gram-negative *E. coli*, a common pathogenic bacterial strain, and a Gram-positive *Corynebacterium glutamicum* (*C. glutamicum*), a widely used bacterial strain in industrial research, is also evaluated. The novelty of this work lies in introducing new avenues for the facile and economic large-scale production of AgNPs, while taking into consideration the high environmental concerns associated with the conventional synthetic routes that require the use of hazardous chemicals.

## 2. Materials and Methods

### 2.1. Microorganisms and Growth Conditions

Cyanobacterium *Pseudanabaena/Limnothrix* sp. was taken from cultural collections of the Department of Biological Applications and Technologies. The standard culture conditions for this work were as follows: 250 mL Erlenmeyer flasks containing 100 mL of BG-11 medium (Fluka Analytical, Sigma-Aldrich, St. Louis, MO, USA) were placed under continuous white LED light intensity of 60 μmol of photons m^−2^ s^−1^ at a temperature of 23 ± 1 °C under shaking at 100 rpm (MRC Refrigerated Shaker Laboratory Incubator, 600 × 480 mm plate, 0–60 °C, 250 rpm) till the stationary phase of culture [14]. Cyanobacterium cells were also cultivated under green, blue, and red LED light intensity of 60 μmol of photons m^−2^ s^−1^ at a temperature of 23 ± 1 °C under shaking at 100 rpm till the stationary phase of culture. In addition, they were cultivated under heterotrophic conditions, in the absence of light, using glucose, 20 g L^−1^ as a carbon source and yeast (1.5 g L^−1^) as a nitrogen source, at a temperature of 23 ± 1 °C under shaking at 100 rpm till the stationary phase of culture.

*E. coli* strain BL21(DE3) and *C. glutamicum* strain ATCC 21253 were taken from cultural collections of the Department of Biological Applications and Technologies. The strains were recovered from cryo-preservation and grown on Luria Bertani (LB) medium (1% tryptone, 1% sodium chloride, 0.5% yeast extract) (Lennox, LAB173, Lancashire, UK) at 37 ± 1 °C under shaking at 180 rpm (MRC Laboratory Shaker Incubator, 450 × 450 mm, 400 rpm, 70 °C). They were stored on LB agar (LAB, a Neogen company, Lancashire, UK) slants at 4 °C [14]. 

### 2.2. Measurement of Phycobiliprotein Concentrations

Phycobiliprotein concentrations were determined at the end of cell culture. In addition, 1 mL of the culture was centrifuged and resuspended in an equivalent volume of deionized water. Phycobiliprotein was extracted by ultrasonication. The debris and intact cells were removed by centrifugation at 4000 rpm (with 16.4 cm rotor radius) for 5 min. The quantitative determination of phycocyanin (PC), allophycocyanin (APC), and phycoerythrin (PE) were performed by measuring the absorbance of the extract at wavelengths of 615 and 652 nm, and using the following equations: PC (mgmL^−1^) = ([OD]_615 − 0.474 [OD]_(652))/5.34;
APC (mgmL^−1^) = ([OD]_562 − 0.208 [OD]_615)/5.09;
PE (mgmL^−1^) = ([OD]_562 − 2.41∗PC − 0.849∗APC)/9.62.

### 2.3. Biosynthesis of Silver Nanoparticles

*Pseudanabaena/Limnothrix* sp. cells at log phase were harvested by centrifugation at 4000 rpm (with 16.4 cm rotor radius) for 5 min, washed twice with deionized water and then were freeze-dried. After that, 100 mg of cells were suspended in 100 mL of water by ultrasonication. The debris and intact cells were removed by centrifugation at 4000 rpm for 5 min, and the cellular extract was filtered (Pall Corporation, 47 mm, pore size 0.45 μm) under vacuum to 4–8 pH, using 1 M HCl solution or 1 M NaOH solution accordingly. The filtered extract was treated with aqueous AgNO_3_ (Sigma-Aldrich, St. Louis, MO, USA) solutions (0.5, 1, and 2 mM) in Erlenmeyer flasks. The reaction mixture was stirred at 20 °C, and the pH was adjusted to 4–8 pH. The extent of reaction was monitored by probing the reduction of Ag^+^ ions to metallic silver (Ag^0^) and silver oxides (Ag_x_O_y_), which corresponded to the formation of AgNPs [15,16]. The resulting AgNPs were collected by centrifugation at 10,000 rpm (with 8.9 cm the rotor radius of the inner ring and 10.1 cm of the outer ring) for 10 min and washed twice with deionized water. The aqueous residues from the nanoparticles were removed by lyophilization. The most suitable conditions were used to produce more AgNPs and for further characterization [17,18,19].

### 2.4. Characterization of Silver Nanoparticles

UV-vis absorption spectra of the nanoparticles as aqueous dispersions were obtained with a JASCO V-730 spectrophotometer (Tokyo, Japan), and were recorded over the wavelength range of 300–800 nm. Powder X-ray diffraction patterns were recorded using a D8 ADVANCE diffractometer (Bruker) with a CuKα radiation (wavelength of 1.5406 Å) and a secondary beam graphite monochromator. The patterns were recorded from 2–80° 2θ at a constant rate of 0.01° s^−1^. Fourier-transform infrared (FTIR) spectra measurements were obtained with an FTIR-8400 spectrometer (Shimadzu, Kyoto, Japan) using KBr pellets. Each spectrum was acquired by averaging 32 scans in the range of 400–4000 cm^−1^. Thermogravimetric (TGA) analysis was performed using a PerkinElmer Pyris Diamond TG/DTA. A sample of approximately 5 mg was heated in air from 25 °C to 850 °C, at a rate of 5 °C min^−1^. X-ray photoelectron spectroscopy (XPS) measurements were performed under ultra-high vacuum at a base pressure of 6 × 10^−9^ mbar using a SPECS GmbH spectrometer equipped with a monochromatic Mg Kα source (photon energy = 1253.6 eV) and a Phoibos-100 hemispherical analyzer (Berlin, Germany). Spectral analysis included a Shirley background subtraction and peak separation using Gaussian-Lorentzian functions in the Winspec least-square fitting program developed at the LISE laboratory, University of Namur, Belgium. Atomic force microscopy (AFM) images that were acquired in tapping mode with a Bruker Multimode 3D Nanoscope (Ted Pella Inc., Redding, CA, USA) using a microfabricated silicon cantilever type TAP-300G, with a tip radius of <10 nm and a force constant of approximately 20–75 N m^−1^. Nanoparticles from aqueous dispersion were deposited onto Si-wafer substrates by drop-casting (~0.01 mg mL^–1^). The Si-wafers (P/Bor, single-side polished, Si-Mat) used in the AFM imaging were cleaned before use for 20 min in an ultrasonic bath (160 W) with water, acetone (≥99.5 % Sigma-Aldrich, St. Louis, MO, USA), and ethanol (≥99.5 % Sigma-Aldrich, St. Louis, MO, USA). Transmission electron microscopy (TEM) images were obtained using an FEI Tecnai 12 BioTwin microscope. The nanoparticles were dispersed in ethanol, and the dispersion was drop-casted onto a lacy copper grid then dried for 1 h prior to the analysis.

### 2.5. Antibacterial Activity

Exponential phase cells of a Gram-negative strain *E. coli* and a Gram-positive strain *C. glutamicum* (~107 CFU mL^−1^ for each strain) were added to aqueous dispersions of AgNPs of different particle concentrations (up to 50 μg mL^−1^) containing 0.9 wt% NaCl. The samples were continuously agitated at 180 rpm (MRC Laboratory Shaker Incubator, 450 × 450 mm, 400 rpm, 70 °C) at a constant temperature of 37 °C for a time period of 12 h. At specific time intervals during this period, 25 μL of each sample were added each time to a Nunclon™ Delta 96-Well MicroWell™ Plate (Thermo Scientific) containing a sterile LB broth medium, and the absorbance value of OD600 was recorded. The lethal effect of AgNPs (LE) was then calculated as the percentage growth of treated cells compared to control at exponential growth phase [20,21,22]. Median lethal concentrations of AgNPs against *E. coli* and *C. glutamicum* were evaluated for each exposure time using the calculator freely available at https://www.aatbio.com/tools/lc50-calculator; accessed on 4 March 2022 (AAT Bioquest, Inc. (5 June 2022). Quest Graph™ LC50 Calculator. AAT Bioquest (Sunnyvale, CA, USA) [23].

## 3. Results and Discussion

### 3.1. Biosynthesis of Silver Nanoparticles

UV-vis spectroscopy was utilized to probe the reduction of Ag^+^ ions associated with the formation of AgNPs. The UV-vis spectra of the as-produced AgNPs (Figure 1) revealed that the synthesis was strongly influenced by the pH of the reaction medium.

The absorption bands at 400–450 nm confirmed the formation of AgNPs [14,24]. Furthermore, the spectra also revealed that the formation of AgNPs were favorable under slightly acidic (pH 5–6) and neutral (pH 7) conditions. Our results are in agreement with the findings of Tomer et al. [25], who reported that pH 7 was the optimum for the formation of AgNPs. Moreover, under slightly basic conditions (pH 8), the SRP peak redshifted to 540 nm, hence implying the formation of AgNPs that are larger in size [16,26]. However, our results are inconsistent with a previous study that reported favorable formation of AgNPs under basic conditions [27]. The effect of AgNO_3_ concentration on the formation of AgNPs was also investigated. At 0.5, 1, and 2 mM AgNO_3_, no differences were observed in either the light spectra or the yield of the reaction measured by the dry weight of the as-synthesized nanoparticles. Therefore, the lowest concentration (0.5 mM AgNO_3_) and neutral conditions (pH 7) were used in the subsequent experiments following other similar works [14,17,27,28].

### 3.2. Effect of Cultivation Light Wavelength on the Synthesis of Silver Nanoparticles

Cyanobacteria are characterized by phycobilisomes, which constitute protein complexes, encored to thylakoid membrane and playing crucial role in the light-harvesting process and the photosynthetic activity. The cyanobacterial cells show characteristic absorption bands in the range of 500 to 700 nm because of the aforementioned pigments. Carotenoids give a characteristic band at ~550 nm, whereas the members of the phycobilisomes family (PC and PE) at 450–660 nm [29,30]. PE of *Pseudanabaena/Limnothrix* sp. cells exhibits an absorption band at 560 nm while PC at 620 nm. As can be seen in Figure 2, the UV-vis spectrum of the initial cyanobacterial extract showed two peaks at 560 and 620 nm corresponding to the PE and PC proteins, both of which disappeared after the synthesis of AgNPs as shown in Figure 1. The disappearance can be attributed to the denaturation of PE and PC during the formation of AgNPs [30]. In contrast, the peak at 675 nm corresponding to chlorophyll-a was present in both of the spectra of the cyanobacterial extract and AgNPs.

As there is evidence for the possibility of phycobilisomes influencing the synthesis of AgNPs, we studied the effect of different wavelengths on the production and composition of pigments during the growth of cyanobacterial cells. The obtained cyanobacteria extracts were then used to synthesize AgNPs under optimum conditions, as described before. When *Pseudanabaena/Limnothrix* sp. cells were grown heterotrophically in the absence of light, their photosynthetic systems were inactive, leaving no measurable amounts of photosynthetic dyes. Although cyanobacterial cells showed high biomass production under heterotrophic conditions, reaching a dry cell weight of approximately 7 g L^−1^, their extract did not lead to the formation of AgNPs, since there was no discoloration of the extract and no SRP band was observed in the UV-vis spectra (Figure 3). Moreover, cyanobacterial cells were not capable of growing satisfactorily under red and blue light. Under these conditions, AgNPs were not produced as well. When cells were cultivated under green LED light, the phycocynanin and PE levels were 73.2 mg/gbiomass and 12.3 mg/gbiomass and the AgNPs production was 4.2 mg, derived from 10 mg cell dry biomass. Finally, cultivation under white light was the most ideal condition, as PC and PE levels were at 152.2 and 32.1 mg/gbiomass, respectively, and the amount of produced AgNPs was 6.2 mg, derived from 10 mg cell dry biomass (Table 1).

### 3.3. Characterization of Silver Nanoparticles

The X-ray diffractogram of the as-produced nanoparticles (Figure 4) showed two peaks at 27.9° and 32.7° corresponding to the (110) and (111) crystal planes of silver oxide, respectively. The peaks centered at 46° and 77.2° can be attributed to (200) and (311) crystal planes of Ag^0^ [31].

The FTIR spectra of *Pseudanabaena/Limnothrix* sp. Cells and the as-synthesized AgNPs (Figure 5) revealed the reducing functional groups characteristic of the biomolecules that took part in the reduction of the Ag^+^ ions [32,33,34]. The peak at 3290 cm^−1^ corresponds to N-H stretching vibrations, which is characteristic of proteins and carbohydrates. The peaks at 3000–2800 cm^−1^ can be attributed to the C-H stretching vibrations of the acyl chains. The peaks at 1662 cm^−1^, 1526 cm^−1^, and 1245 cm^−1^ correspond to the amide I, II, and III vibrational modes of peptide bonds, respectively, and result from C=O stretching, N-H bending, C-N stretching vibrations [32,33]. The bands at 950–1200 cm^−1^ can be attributed to the stretching vibrations of the epoxide (C-O-C) groups [33]. Last but extremely important is the presence of bands in the range of 900–600 cm^−1^ that are associated with the NH_2_ region of amines and the bands around 600 cm^−1^ are a confirmation of AgNPs formation [35]. The binding of protein with AgNPs constitutes a confirmation of the stability of produced NPs [36]. Moreover, the FTIR spectra showed relative shifts in the peak positions, indicating the role of cyanobacterium extract to AgNPs formation. Our results are in agreement with those of El-Naggar et al. and Hamouda et al., who revealed that biomolecules from cell extracts may contribute to the reduction of Ag^+^ ions [37,38]. Together, the XRD and FTIR results confirm the formation of AgNPs and the existence of residual capping compounds.

TGA was utilized to determine the silver content in the as-produced AgNPs. The TGA thermogram of AgNPs (Figure 6) showed a mass loss of ~4% below 120 °C, which is probably due to the absorbed water molecules. Between 120 and 500 °C, a mass loss of 66% was observed, which can be attributed to the decomposition of the organic capping agents present on the surface of the AgNPs. The decomposition of the organic species proceeded until 900 °C with an additional mass loss of ~10%. The remaining 20% corresponds to the silver mass content in AgNPs.

Figure 7a shows the XPS survey of AgNPs. The survey revealed the presence of carbon, oxygen, silver, and nitrogen species with traces of silicon from the substrate onto which the sample was deposited. The deconvoluted high resolution spectrum of Ag 3d (Figure 7b) revealed the following three components: Ag_2_O (4.2%), AgO (58.9%), and Ag^0^ (36.9%). The deconvoluted core-level C 1s spectrum showed five components giving significant information for the carbon functionalities surrounding the AgNPs as presented in detail in Figure 7c. Finally, the N 1s spectrum was fitted into two peaks; one corresponding to the NH_2_ groups at lower binding energy (399.9 eV) and the second to the N-H groups. The atomic composition of the sample is shown in Table 2.

The morphology of the as-synthesized AgNPs nanoparticles was examined by AFM. Representative AFM images of AgNPs deposited on Si-wafer are presented in Figure 8a–c. In addition, the height profile (Figure 8d) revealed nanoparticles with an average size of 6–7 nm, which is in accordance with previous studies [39,40,41,42]. Interestingly, the AFM images also showed that our AgNPs were not only ultra-small, but also exhibited high monodispersity, something that was not observed in previous reports. For instance, the size of AgNPs synthesized using other bacterial strains was as follows: *Microcoleus* sp. (44–79 nm); *Synechococcus* sp. (15.2–266.7 nm); *Anabaena doliolum* (10–50 nm); and *Leptolyngbya* sp. (20–35 nm) [17,39,40,41].

The morphological characteristics of the derived AgNPs were examined using electron microscopy. TEM images (Figure 9) revealed the presence of nanoparticles exhibiting shape regularity and size uniformity, which is in agreement with the AFM images (Figure 8).

### 3.4. Antibacterial Properties of Silver Nanoparticles

The bactericidal activity of AgNPs, produced using the *Pseudanabaena/Limnothrix* sp. cellular extract, was studied against Gram-negative *E. coli* and Gram-positive *C. glutamicum* at various concentrations of AgNPs ranging from 2 to 50 μg mL^−1^ and for various interaction times (0.5 to 12 h). The AgNPs exhibited enhanced bactericidal properties against Gram-positive and Gram-negative bacterial cells at low concentrations. Dosages over 25 μg mL^−1^ led to cell deaths of more than 95% of bacterial cells for both strains, when the interaction time was more than 2 h (Figure 10). The antibacterial activity of AgNPs is dose- and time-dependent. The AgNPs dose and the time of their interaction with the bacteria act synergistically: for smaller doses, more time was needed to achieve the same result. When this dose was decreased to 15 μg mL^−1^, both bacterial strains needed 4 h of interaction for over 95% cell population suppression. Moreover, a significant lethal effect on *E. coli* cells of about 70% was observed for the 10 μg mL^−1^dose after 12 h of interaction. The same effect, a 70% lethal effect, was observed for the *C. glutamicum* cells for the 20 and 50 μg mL^−1^ doses after 2 and 1 h of interaction, respectively. It is noteworthy that increasing the dose of AgNPs to 50 μg mL^−1^ can lead to the death of approximately 50% of *C. glutamicum* cells after just 30 min of exposure. Previous works related to the antibacterial properties of AgNPs reported 50 μg mL^−1^AgNPs as the minimum concentration able to cause 100% lethal effect of bacterial cells [35,43,44]. Our calculations show that the amount of Ag^+^ ions interacting with the cells was approximately 0.2 μg Ag^+^/μg AgNPs. The sublethal concentrations of Ag^+^ ions have been reported at the range of 3 to 9 μg mL^−1^ [44].

The lethal concentration (LC_50_) is defined here as the nanoparticle concentration which causes the death of 50% of the bacterial population. For our AgNPs, the LC_50_ values are presented in Table 3 for the aforementioned interaction times. For all the times of interaction, the LC_50_ values are lower for *C. glutamicum* cells than for *E. coli* cells. The extremely low LC_50_ values reveal the remarkably high antibacterial activity of our AgNPs. The differences among bacterial strains are probably due to the differences between the cellular membranes Gram-positive and Gram-negative bacteria, as the layer of peptydoglycans and the periplasmic space make the Gram-negative bacterial strains more resistant to antibiotics and bactericidal agents [13].

Overall, AgNPs produced using *Pseudanabaena/Limnothrix* sp. cells extracts exhibited enhanced bactericidal activity against Gram-positive and Gram-negative bacterial cells at low concentrations and in a short period of time.

## 4. Conclusions

The present work demonstrates for the first time the use of cyanobacterium *Pseudanabaena/Limnothrix* sp. cells extract as the reductant and capping agent for the synthesis of extra-small AgNPs. The synthetic process can be characterized as rapid, cost-efficient, and eco-friendly. The cyanobacterium extract act as the reducing and capping agent, and the use of AgNO_3_ was minimized. The characterization of the synthesized AgNPs was carried out through multiple spectroscopic and microscopic techniques, revealing monodisperse spherical nanoparticles with a size of 6–7nm and 20% Ag content, while we assume a first impression about the possible contribution of phycobilisomes in the synthetic process and the biological compounds that act as reducing and capping factors. The produced biogenic AgNPs are environmentally safer compared to them produced with classic chemical and physical methods, as the synthetic conditions are gentle, with no energy consumption—the synthesis achieved in room temperature of 20 °C, and there is an absence of toxic compounds. Moreover, through the use of *Pseudanabaena/Limnothrix* sp. cells, a new perspective for the overburdened eutrophic cyanobacterium habitant is offered. Moreover, the strong bactericidal properties that AgNPs have against both Gram-positive and Gram-negative bacterial strains make them as a promising antibacterial tool that can be exploited in several applications. Concluding in this study, we propose a green synthesis of AgNPs from a new cyanobacterium strain. The use of microorganisms as bioreactors gives a new aspect in the field of nanotechnology, as it is capable of replacing the traditional physical or chemical synthetic processes. In contrast to other works, in this study, we utilized cyanobacterial cellular extracts to produce ultra-small monodisperse nanoparticles exhibiting a high degree of shape uniformity as demonstrated by AFM and TEM. Moreover, the novelty of this work is based on the fact that the conditions are extremely mild compared to similar works, and this may be the reason for the presence of multiple phases, such as Ag^0^, Ag_2_O, and AgO. Last but not least, this work proposes a novel biosynthesis of a strong antibacterial agent that takes into consideration the environment, and lays the foundation for future research toward exploring the mechanism behind this process for the potential large-scale synthesis of AgNPs. 

## Figures and Tables

**Figure 1 nanomaterials-12-02296-f001:**
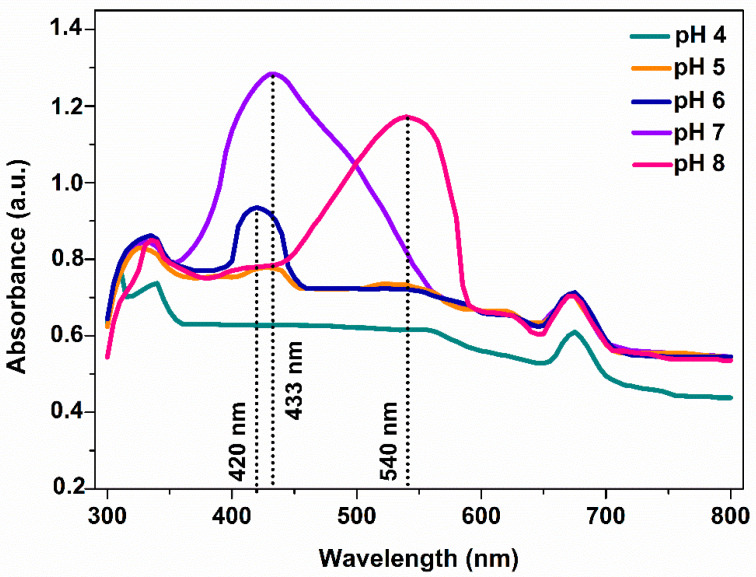
UV-vis spectra of AgNPs synthesized at different pH values.

**Figure 2 nanomaterials-12-02296-f002:**
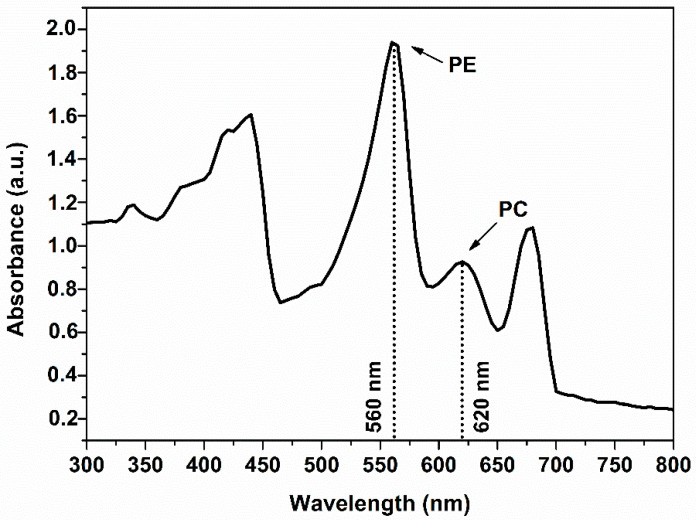
UV-vis spectra of cyanobacterium extract, where the absorption bands corresponding toPE and PC are indicated.

**Figure 3 nanomaterials-12-02296-f003:**
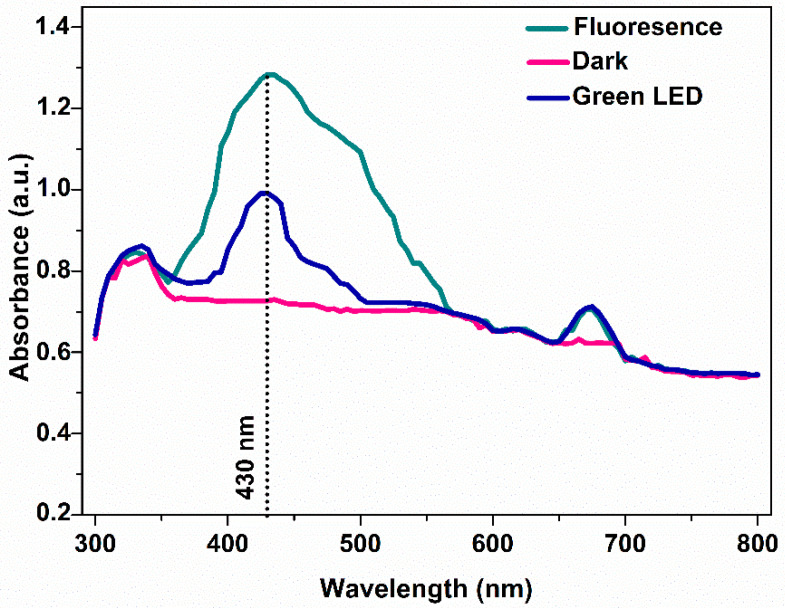
UV-vis spectra of produced AgNPs under different cultivation light conditions.

**Figure 4 nanomaterials-12-02296-f004:**
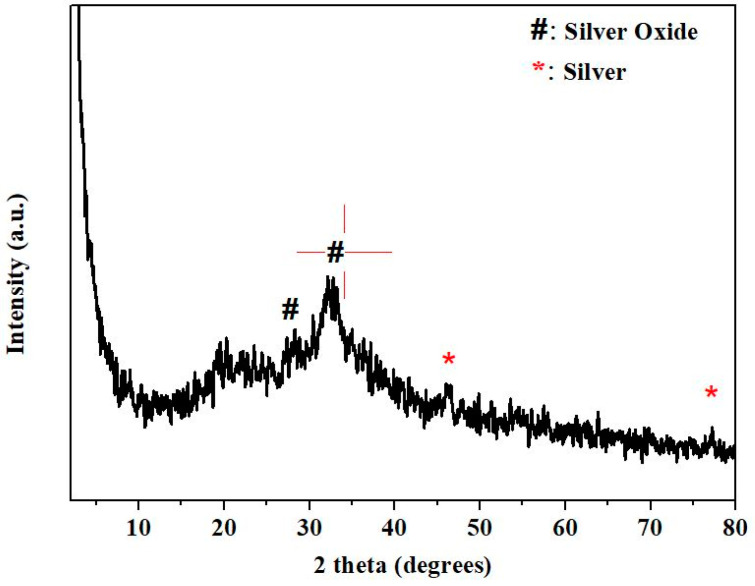
X-ray diffraction of the as-synthesized AgNPs.

**Figure 5 nanomaterials-12-02296-f005:**
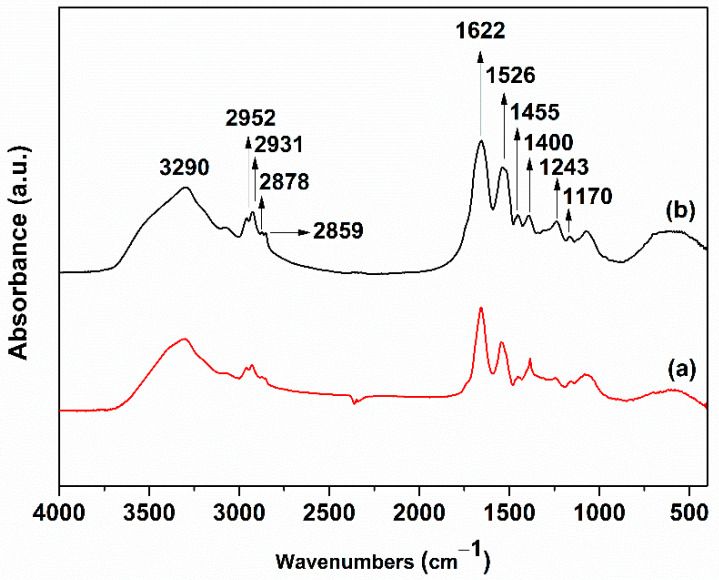
FTIR spectra of (**a**) *Pseudanabaena/Limnothrix* sp. cells and (**b**) synthesized AgNPs. Each peak in the spectrum of the synthesized AgNPs represents a specific molecular vibration. The corresponding functional groups are shown in the spectra.

**Figure 6 nanomaterials-12-02296-f006:**
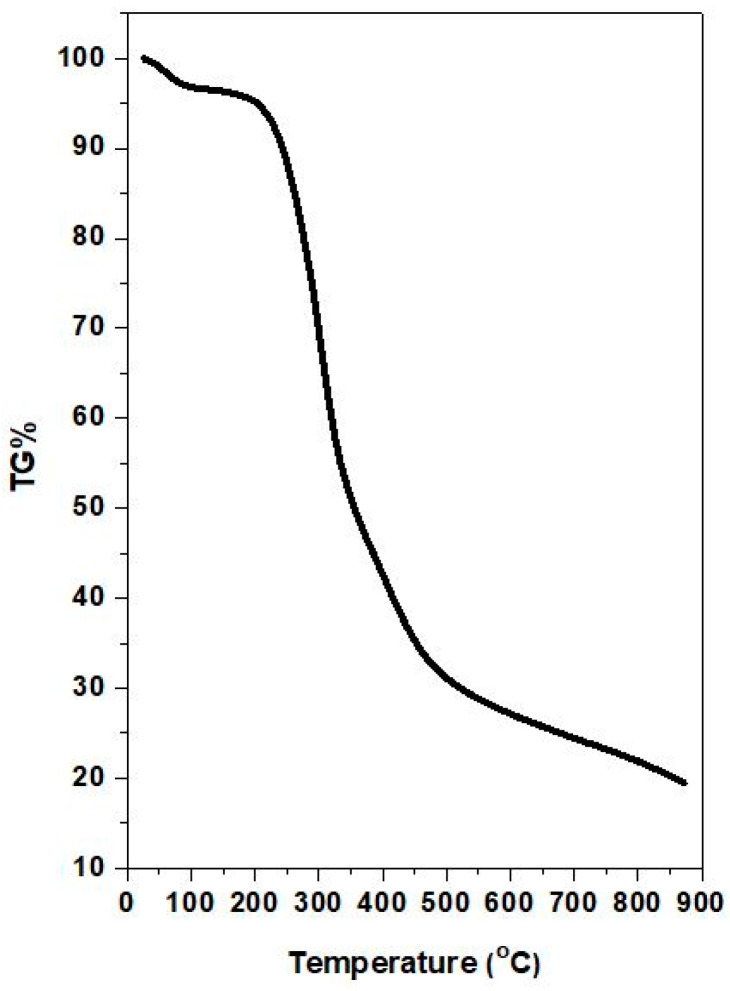
TGA thermogram of the produced AgNPs.

**Figure 7 nanomaterials-12-02296-f007:**
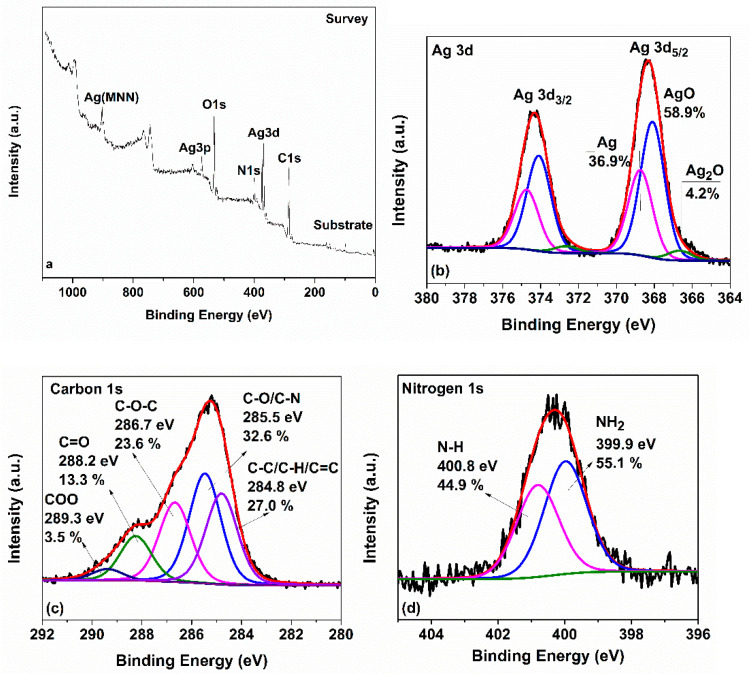
(**a**) XPS survey of the produced AgNPs; (**b**) Ag 3d; (**c**) C 1s; and (**d**) N 1s X-ray photoelectron spectra of AgNPs.

**Figure 8 nanomaterials-12-02296-f008:**
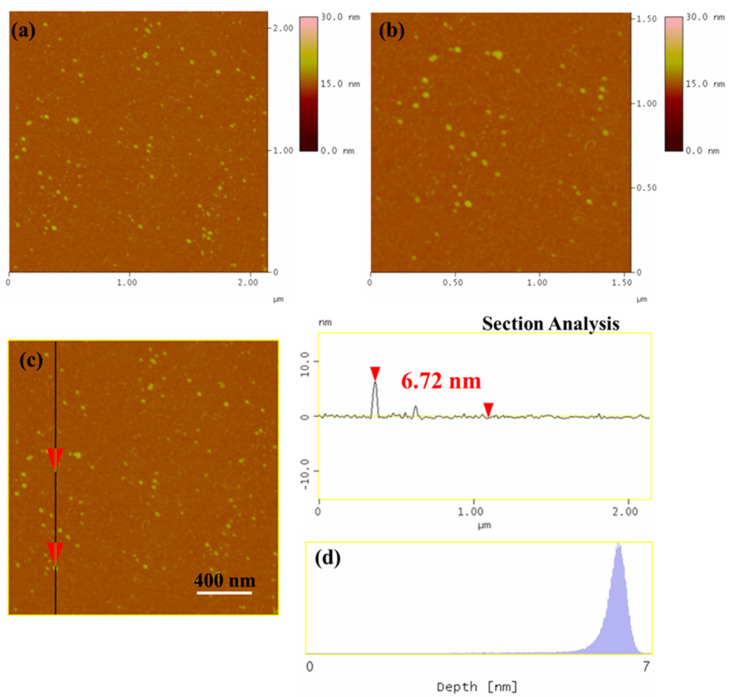
(**a**,**b**) AFM height images, (**c**) cross sectional analysis line, and (**d**) particle size distribution of AgNPs deposited on a Si-wafer.

**Figure 9 nanomaterials-12-02296-f009:**
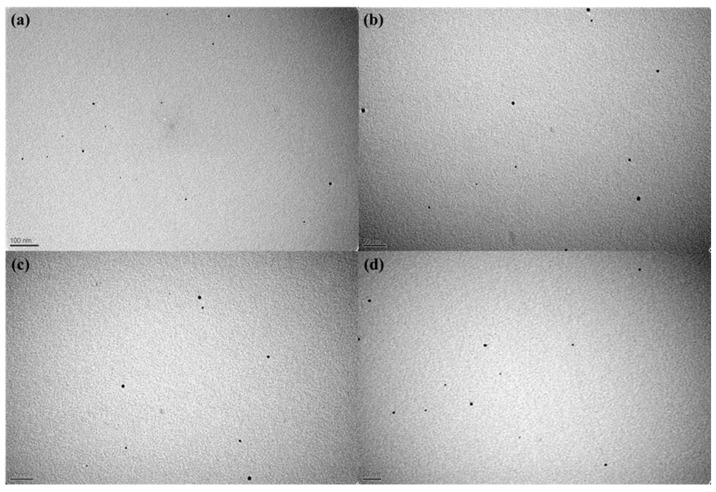
TEM images of Ag NPs taken from different regions(**a**–**d**).

**Figure 10 nanomaterials-12-02296-f010:**
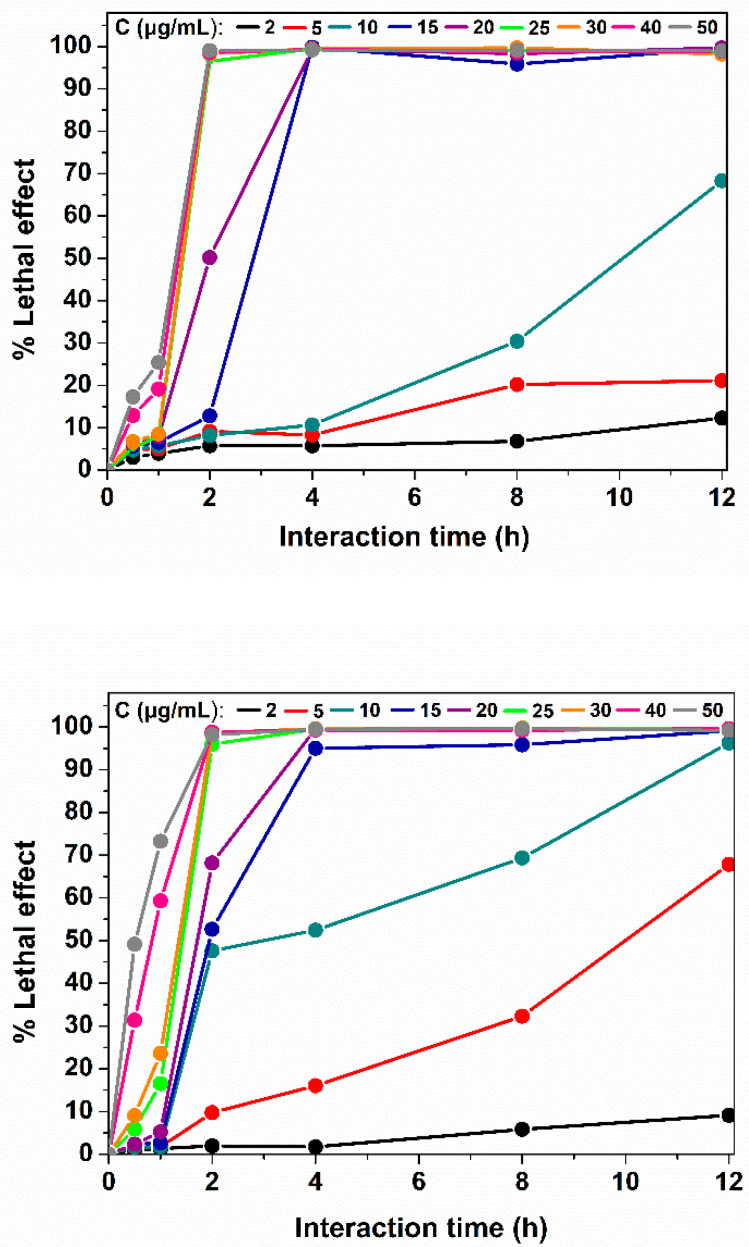
Lethal effect of synthesized AgNPs on *E. coli* (**top**) and *C. glutamicum* (**bottom**) cells under different concentrations and interaction times. All measurements were in triplicates while the standard deviation is represented.

**Table 1 nanomaterials-12-02296-t001:** Effect of light wavelength on phycocyanin and PE concentrations on *Pseudanabaena/Limnothrix* sp. cells and AgNPs synthesis.

LED Light	AgNPsDry Weight (mg)	PC (mg/g Biomass)	PE (mg/g Biomass)
White	6.2 ± 0.1	152.2 ± 1.0	32.1± 0.6
Green	4.2 ± 0.2	73.2 ± 0.7	12.3 ± 0.7
Blue	nd	<0.1	<0.1
Red	nd	<0.1	<0.1
Dark	nd	nd	nd

These results are in agreement with previous studies, which reported that PC can be used for the synthesis of spherical and elongated AgNPs [15,16].

**Table 2 nanomaterials-12-02296-t002:** Surface elemental composition of AgNPs.

Element	Percentage%	Error%
C	64.1	3.1
Ag	21.4	1.3
O	6.4	0.4
N	8.1	0.5

**Table 3 nanomaterials-12-02296-t003:** LC_50_ of AgNPs against *E. coli* and *C. glutamicum* at various interaction times.

Interaction Time (h)	LC_50_ (μg mL^−1^)	
	*E. coli*	*C. glutamicum*
0.5	nc	52.0 ± 0.4
1.0	>50	36.0 ± 1.0
2.0	19.5 ± 0.5	14.5 ± 0.5
4.0	14.0 ± 0.5	9.6 ± 0.4
8.0	10.5 ± 0.4	7.5 ± 0.4
12.0	7.2 ± 0.3	4.5 ± 0.4

nc: not calculated.

## Data Availability

Not applicable.

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
