# Peer review of "Green Synthesis and Characterization of Silver Nanoparticles with High Antibacterial Activity Using Cell Extracts of Cyanobacterium Pseudanabaena/Limnothrix sp."

_nanomaterials, 2022, doi:10.3390/nano12132296_

Round 1

Reviewer 1 Report

The manuscript entitled "Green synthesis of silver nanoparticles with high antibacterial activity using cell extracts of cyanobacterium Pseudanabaena Limnothrix sp." publishes the results of a study on the green synthesis of silver nanoparticles using a new object – cyanobacteria, which is consistent with existing scientific trends. The article is written in good language, the narrative is logical, the conclusions are confirmed by the results. It is proposed to accept the article after minor changes.

1) Lines 83-86. The novelty of this work lies in the use of a new object for green synthesis. As for the new possibilities of industrial synthesis, this topic is touched upon in the article no more than in dozens of other works devoted to the biogenic synthesis of AgNPs.

2) The authors could add information about the bactericidal and fungicidal action of biogenic silver nanoparticles, for example, https://doi.org/10.3390/mi1212148 .

3) Pages 8 and 10. There is too much free space at the bottom of the pages.

4) Figure 7 is separated by a table. The signatures are made in the usual font and italics, that is, without uniformity.

5) Figure 9. It is not clear why 4 fields of view were taken, one would be enough. The concentration of nanoparticles is small, the increase is not enough.

6) Figure 10 is missing.

7) Figure 11. The description is quite difficult to read. The signatures of the curves could be made in the figure, or the description could be rewritten to make it clearer.

Author Response

Referee: 1

  1. Lines 83-86. The novelty of this work lies in the use of a new object for green synthesis. As for the new possibilities of industrial synthesis, this topic is touched upon in the article no more than in dozens of other works devoted to the biogenic synthesis of AgNPs.

We partly agree with the reviewer. We may use the same green synthetic process, but the results are more than improved compared in previous reports as we have mentioned in the manuscript. For instance, we have minimized the AgNO3 concentration, we have used only mild conditions (200C and pH 7). At last, the produced NPs are of extra small dimensions and of great homogeneity – compared to previous produced AgNPs, resulting in increase of their effectiveness and the possible applications.  

  1. The authors could add information about the bactericidal and fungicidal action of biogenic silver nanoparticles, for example, https://doi.org/10.3390/mi12121480 .

We thank the reviewer for this suggestion and some additional information about the bactericidal action of biogenic silver nanoparticles has been added in page 1and line 39-41. “Moreover, it is worth mentioned the fact that biogenic silver nanoparticles present greater effectiveness compared to classical antibiotics as microbes develop much lower resistance to them.”Additionally ref https://doi.org/10.3390/mi12121480 has been added in line 41 according to reviewers suggestion

  1. Pages 8 and 10. There is too much free space at the bottom of the pages.

 We modified the draft and there is not much free space any more

  1. Figure 7 is separated by a table. The signatures are made in the usual font and italics, that is, without uniformity

The table has been modified uniformly in the text according to reviewers’ suggestion

  1.  Figure 9. It is not clear why 4 fields of view were taken, one would be enough. The concentration of nanoparticles is small, the increase is not enough.

The reason we place 4 different images for TEM was because we wanted to show the homogeneity of Ag nanoparticles concerning their shape and size. The images represent four different scans in four different areas of the samples. We would like to keep them unless the editor has different opinion.

  1. Figure 10 is missing.

We apologize for the numeration. Figure 11 has been replaced with figure 10 in the text. There are 10 figures in the draft.

  1. Figure 11. The description is quite difficult to read. The signatures of the curves could be made in the figure, or the description could be rewritten to make it clearer.

 We modified the figure according to reviewer suggestion to make it clearer.

Reviewer 2 Report

In my opinion this is interesting manuscript and it should be published after the correction of  indicated issues

-        The process of preparation of nanostructures is not reproducible. The authors should indicate the volume of bacterial extract used during the preparation process. The acid used for the pH adjustment should be described.

-        It is misleading that the authors use diverse definitions to describe products of synthesis. In the abstract one can find information about “silver nanoparticles in the form of Ag0 and AgxO”, then they are describe “metallic silver Ag0 and AgO and formed the nanoparticles” Please correct this issue

-        Based on the extinction spectra it is difficult to indicate that plasmonic structures of silver nanoparticles were obtained in each preparation attempt. Please discuss the lack of characteristic absorption bands in the spectra. Please improve the Figures – dots should be used

-        The sample preparation for AFM imaging should be described in detail. In my opinion the TEM micrographs are not good. I suggest adding micrographs presenting more nanoparticles. Based on TEM micrographs and AFM images please prepare size distribution of nanoparticles. The authors should present the average size of nanoparticles and polydispersity index.

-        The experimental methods used for the determination of nanoparticle concentration in the suspension should be described.

-        In my opinion the stability of nanostructures dispersed in biological media should be determined.

-        The authors should compare the effectiveness of obtained nanostructures with other literature reports. Please discuss benefits and merits of nanoparticles obtained according to the developed method.

Author Response

Referee: 2

  1. The process of preparation of nanostructures is not reproducible. The authors should indicate the volume of bacterial extract used during the preparation process. The acid used for the pH adjustment should be described.

We thank the reviewer for this comment. We added the pH conditions in materials and methods parts and more specific in lines 117, 118 and 124, more details about the process.

“and then were freeze-dried. After that, 100mg of cells were suspended in 100ml of water by ultrasonication.” (Line 117-118)

“to 4-8 pH, using 1M HCl solution or 1M NaOH solution accordingly” (Line 121)

  1. It is misleading that the authors use diverse definitions to describe products of synthesis. In the abstract one can find information about “silver nanoparticles in the form of Ag0 and AgxO”, then they are describe “metallic silver Ag0 and AgO and formed the nanoparticles” Please correct this issue

We would like to thank the reviewer for his comment. In the title “Green synthesis of silver nanoparticles with high antibacterial activity using cell extracts of cyanobacterium Pseudanabaena/Limnothrix sp.” We refer to Ag Nanoparticles meaning that the final nanoparticles are in different states (Ag0, AgO, Ag2O). In the abstract we discuss in detail that the possible states of Ag nanoparticles are metallic silver and AgxOy. With the abbreviation AgxOy we refer to AgO and Ag2O. We leave it to editor decision if we need to delete AgxOy and each time to state AgO and Ag2O.

  1. Based on the extinction spectra it is difficult to indicate that plasmonic structures of silver nanoparticles were obtained in each preparation attempt. Please discuss the lack of characteristic absorption bands in the spectra. Please improve the Figures – dots should be used

I agree. Please check my comment in the MS. The dots are not making the Graphs better so we will keep the lines.

  1. The sample preparation for AFM imaging should be described in detail. In my opinion the TEM micrographs are not good. I suggest adding micrographs presenting more nanoparticles. Based on TEM micrographs and AFM images please prepare size distribution of nanoparticles. The authors should present the average size of nanoparticles and polydispersity index.

We would like to thank the reviewer for the constructive comment. We added in the draft a more detailed sample preparation for the AFM.

“Atomic force microscopy (AFM) images that were collected in tapping mode with a Bruker Multimode 3D Nanoscope (Ted Pella Inc., Redding, CA, USA) using a microfabricated silicon cantilever type TAP-300G, with a tip radius of <10 nm and a force constant of approximately 20–75 N m−1. Nanoparticles from aqueous dispersion were deposited onto the Si-wafer substrates by drop-casting (∼0.01 mg mL–1). The Si wafers (P/Bor, single-side polished, Si-Mat) used in the AFM imaging were cleaned before use for 20 min in an ultrasonic bath (160 W) with water, acetone (≥ 99.5 % Sigma-Aldrich, St. Louis, MO, USA), and ethanol (≥ 99.5 % Sigma-Aldrich, St. Louis, MO, USA).” (Line 145-153)

The size distribution of AFM is in the MS

Unfortunately, the limited time for review is not enough for detailed TEM analysis but is on our future plans a detailed TEM investigation presenting the average size of nanoparticles (more than 200 nanoparticles in different areas) and polydispersity index.  

  1. The experimental methods used for the determination of nanoparticle concentration in the suspension should be described.

As we have reported in paragraph 2.3, the produced AgNPs were collected by centrifugation and the aqueous residues were removed by lyophilization. So, we don’t talk about nanoparticles solutions. The mass of the produced AgNPs was determined from their weight.

  1. In my opinion the stability of nanostructures dispersed in biological media should be determined.

We would like to thank the reviewer for this comment. In the current study, we work on the optimization of the production of biogenic AgNPs. Additionally, we tested their antibacterial properties so to explore their usability as antibacterial tools with exceptional results. During this investigation we observe that the antibacterial properties of the nanoparticles even after 6 months remained the same, and this is a very strong indication that the stability of the nanoparticles remained the same.

This study constitutes a part of a bigger project and it’s on our future plans to test the stability of the produced AgNPs on different biological media. This will give functionality of the synthesized nanoparticles in diverse biological applications. Our target is to create a multifunctional material and stability is on of our priorities to achieve that goal. 

  1. The authors should compare the effectiveness of obtained nanostructures with other literature reports. Please discuss benefits and merits of nanoparticles obtained according to the developed method.

We thank the reviewer for this comment. We added comparisons with previous literature reports in Results and Discussion part, while we also enhance the Conclusion part, where we talk also about the benefits and merits of the produced biogenic AgNPs. See Referee 5 and question 1.

Reviewer 3 Report

In the manuscript, the authors demonstrated the ability of the cyanobacterium Pseudanabaena/Limnothrix sp. to produce silver nanoparticles. And the antibacterial activity of the silver nanoparticles was investigated. However, the innovation of this manuscript was not enough and the workload was not sufficient to verify the innovativeness. So this manuscript was not enough to be published in Nanomaterials. Meanwhile, the design of some experiments in this manuscript was unreasonable. The main questions are listed below:

1The ability of Phycocyanin and Phycoerythrin to produce silver nanoparticles have been reported in articles published in Colloids and Surfaces B: Biointerfaces (DOI:10.1016/j.colsurfb.2020.111211) and Scientific Reports(DOI: 10.1038 / s41598-018-27276-6). In this manuscript, the authors only investigated the phycocyanin and phycoerythrin of Pamv7 can produce Ag0.Compared with the reported articles, this manuscript was not improved. Hence, the manuscript was less innovative for publishing in Nanomaterials.

2In the manuscript, the antibacterial activity of the silver nanoparticles against Gram-positive Corynebacterium glutamicum bacterial cells was evaluated. However, Corynebacterium glutamicum bacterial cells is widely used in industrial production (DOI: 10.1016/j.ymben.2021.06.011), and is not suitable for antibacterial research. It is suggested using gram-positive bacteria which is commonly used in antibacterial research, such as Staphylococcus aureus.

3In Figure 1, authors described “The absorption bands at 400-450 nm indicate the presence of Ag”. However, the cited reference 13 described “The differences in SPR values may have been due to the different species of cyanobacteria used to produce SNPs”, which was not enough to support the description in this manuscript. Hence, it is difficult to prove that the absorption band at 400-450 nm was the silver nanoparticles. Meanwhile, the SRP peak redshifted to 540nm under pH8 did not explain the produced larger nanoparticles. It is suggested using SEM to verify the produced silver nanoparticles.

4In Figure 4, authors described “The X-ray diffraction of the as-produced nanoparticles (Figure 4) shows two peaks at 27.9o and 32.7o corresponding to the (110) and (111) crystal planes of silver oxide, respectively. The peaks centered at 46o and 77.2o can be attributed to (200) and (311) crystal planes of Ag0”. However, the character peaks of above mentioned were not obvious except for the peak of 32.7o. So, Figure 4 is insufficient to certify the synthesis of silver nanoparticles, please resupply the X-ray spectrum.

5In Figure 5, authors described that the binding of protein with AgNPs constitutes a confirmation of the stability of produced NPs, but no data to certify the stability of AgNPs, please add the relevant data.

6In Figure 11, authors verified the antibacterial activity of silver nanoparticles against E.coli and C. glutamicum at various concentrations. However, in this manuscript, authors did not set up the positive group (such as penicillin, etc.) and control group to verify antibacterial activity of nanoparticles, please supply the relevant experimental data.

7Biosafety is a necessary condition for the application of any antibacterial material in vivo. In this manuscript, in virto safety of nanoparticles was not studied, please add relevant experiments.

8It is suggested to mark different colors in the Figure 1 and 3 in the form of icons.

9It is recommended that the English in the manuscript need to be modified.

Author Response

Referee: 3

In the manuscript, the authors demonstrated the ability of the cyanobacterium Pseudanabaena/Limnothrix sp. to produce silver nanoparticles. And the antibacterial activity of the silver nanoparticles was investigated. However, the innovation of this manuscript was not enough, and the workload was not sufficient to verify the innovativeness. So, this manuscript was not enough to be published in Nanomaterials. Meanwhile, the design of some experiments in this manuscript was unreasonable. The main questions are listed below:

  1. The ability of Phycocyanin and Phycoerythrin to produce silver nanoparticles have been reported in articles published in Colloids and Surfaces B: Biointerfaces (DOI:10.1016/j.colsurfb.2020.111211) and Scientific Reports (DOI: 10.1038 / s41598-018-27276-6). In this manuscript, the authors only investigated the phycocyanin and phycoerythrin of Pamv7 can produce Ag0. Compared with the reported articles, this manuscript was not improved. Hence, the manuscript was less innovative for publishing in Nanomaterials.

The aim of this work was to synthesize biogenic AgNPs, using the extract of a cyanobacterium strain, isolated from an eutrophic lake of our region. Our novelty lies in the fact that these cells yield a dual environmental benefit; the alleviation of an overburdened ecosystem and the production of a high-value product. Continuing, we would like to mention that we used a green ecofriendly synthetic process, where just 0.5mM of AgNO3, in neutral pH and temperature of 20oC were selected as the optimal for our synthesis, compared with the above mentioned papers where extremely alkali pH values, high temperatures (40oC) and AgNO3 concentrations of 10-20mM (which means 20-40 times greater than ours) were used. So, we used milder, less costly and more eco-friendly conditions. Moreover, we produced spherical AgNPs 7-13nm, compared to the produced 7-26nm nanoparticles of the reported work, revealing greater homogeneity. Last but not least, in the reported works, isolated phycoerythrin and phycocyanin were used for the nanoparticles synthesis, compared to our work where we only indicate the contribution of the phycobilisomes family. So, for the above reasons, we still support the novelty of our study.

  1. In the manuscript, the antibacterial activity of the silver nanoparticles against Gram-positive Corynebacterium glutamicum bacterial cells was evaluated. However, Corynebacterium glutamicum bacterial cells is widely used in industrial production (DOI: 10.1016/j.ymben.2021.06.011) and is not suitable for antibacterial research. It is suggested using gram-positive bacteria which is commonly used in antibacterial research, such as Staphylococcus aureus.

The rationale for using the two strains of bacteria has been reported on page 2 and lines 79-82. We wanted to test the possible antibacterial properties of nanomaterials on two pathogenic bacteria, that consist at the same time characteristic examples of the two main bacterial categories (gram- and gram+).  Of course, there are many different genus and strains of bacteria, both gram+/-, but the aim of this work was the production of biogenic AgNPs through a green process and the test of possible antibacterial properties, making them appropriate candidates for many applications. Of course, we could make a screening of the nanoparticles effect on many bacterial strains, and it constitutes a future perspective.

  1. Reference in Figure 1, authors described “The absorption bands at 400-450 nm indicate the presence of Ag”. However, the cited reference 13 described “The differences in SPR values may have been due to the different species of cyanobacteria used to produce SNPs”, which was not enough to support the description in this manuscript. Hence, it is difficult to prove that the absorption band at 400-450 nm was the silver nanoparticles. Meanwhile, the SRP peak redshifted to 540nm under pH8 did not explain the produced larger nanoparticles. It is suggested using SEM to verify the produced silver nanoparticles.

I agree please check my comment in the MS. (Lines 173-194)

  1. In Figure 4, authors described “The X-ray diffraction of the as-produced nanoparticles (Figure 4) shows two peaks at 27.9and 32.7corresponding to the (110) and (111) crystal planes of silver oxide, respectively. The peaks centered at 46and 77.2can be attributed to (200) and (311) crystal planes of Ag0”. However, the character peaks of above mentioned were not obvious except for the peak of 32.7o. So, Figure 4 is insufficient to certify the synthesis of silver nanoparticles, please resupply the X-ray spectrum.

In the draft we mention that we have the production of extremely small (7-13nm) spherical Ag nanoparticles with high homogeneity. The XRD diffraction pattern display peaks which are not obvious for two main reasons that act synergistically. The first reason is that, as we demonstrate from XPS and XRD characterization, there is amorphous carbon enclose the spheres of some nm thickness. The second reason is that due to the very small size of the Ag nanoparticles and the very good distribution (not aggregates) it is difficult to see clear and sharp diffraction peaks. We have many examples from literature that reducing the size of the nm scale for synthesized nanoparticles  (doi: 10.2147/IJN.S36786),(doi:10.1088/0957-4484/16/12/015), (doi: 10.1039/C2CS15325D). So we believe that this diagram enhance the fact that we synthesize ultrafine Ag nanoparticles.

  1. In Figure 5, authors described that the binding of protein with AgNPs constitutes a confirmation of the stability of produced NPs, but no data to certify the stability of AgNPs, please add the relevant data.

We would like to thank the reviewer for this comment. Figure 5 constitutes the FT-IR analysis of the produced AgNPs, enhanced in the new draft with the FT-IR of cyanobacterial cells. So, the referred sentence is about the explanation of figure 5. What we mentioned is the fact that the presence of protein bands at AgNPs, something indicating the possibility that nanoparticles are bound to proteins through free amine groups. The interaction of Ag surface with protein compounds can create more stable AgNPs, preventing them from agglomeration and making them more stable in medium and the role of proteins on AgNPs have been already reported. The absence of agglomeration that may be related with protein binding has been indicated in out work through the characterization techniques. The stability of the produced AgNPs has been determined via a repeat of the characterization process after 6 months but is constitute part of the ongoing project. In this work, we would like to emphasize on the synthetic process.

  1. In Figure 11, authors verified the antibacterial activity of silver nanoparticles against E.coli and C. glutamicum at various concentrations. However, in this manuscript, authors did not set up the positive group (such as penicillin, etc.) and control group to verify antibacterial activity of nanoparticles, please supply the relevant experimental data.

In the current work and specifically in the part of the antibacterial study of the synthesized Ag NPs, we tested the effect of both dose and interaction time of AgNPs with two different bacterial strains, a gram+ and a gram-.  As it is already known, the search for new antibacterial composites is urgent as the resistance of pathogenic bacterial strains is grown rapidly. So, in the current work, we produced metal nanoparticles with antibacterial properties, and it is proven through the antibacterial tests. The aim of the work was neither the comparison with classic and wide-used antibiotics like penicillin, nor the production of a penicil competitor. This work constitutes the first part, related with the synthetic process of biogenic AgNPs with bactericidal properties, and not a classic toxicity study.  

  1. Biosafety is a necessary condition for the application of any antibacterial material in vivo. In this manuscript, in virto safety of nanoparticles was not studied, please add relevant experiments.

As it is previously mentioned, the aim of this work was the green synthesis of silver nanoparticles from cyanobacterial cells. Moreover, a first antibacterial test of the produced NPs towards a gram+ and a gram- bacterial strain took place, indicating the great bactericidal properties of the produced nanoparticles, making them possible candidates as nanosupports in various environmental, biological and medical applications. We would like here to mention that silver nanoparticles have already found fertile place in biomedicine due to their unique physical and chemical properties as well the biosafety which have been extensively studied to organs and systems in vitro/and vivo (doi: 10.7150/thno.45413) (doi: 10.1159/000093928). However, we totally agree with the reviewer and we would like to express our thanks for this comment, saying that biosafety tests is known that constitute a great and necessary study before the application of the produced nanoparticles. That’s why we mention out that Ag NPs are potential candidates for biomedical applications, satisfying the bactericidal criteria and organizing the biosafety tests as part of the ongoing work.

  1. It is suggested to mark different colors in the Figure 1 and 3 in the form of icons.

We modified Figure 1 and Figure 3 according to reviewer suggestion

  1. It is recommended that the English in the manuscript need to be modified.

We thank the reviewer for his enquiry to improve grammatically the draft. The draft has been carefully checked and many points have been improved in the draft emphasizing in conclusions and abstract. All changes are in red font.

Reviewer 4 Report

In this paper, extra-small silver nanoparticles were prepared using Pseudanabaena/Limnothrix sp cells extract and their characteristics were analyzed using FTIR, XPS, AFM and TEM. Antibacterial tests were performed on E. coli and C. glutamicum to evaluate the final biological function.

The standards for the quality of research are clearly lacking.

Authors recommend checking for grammatical and syntax errors.

There are errors in the introduction and discussion, and many statements are cited as documents that are not representative. You should use representative articles or high-impact-factor reviews, not random articles from random journals. References are available, but the introduction or discussion sections lack much description of the use of Pseudanabaena/Limnothrix sp cells extract .

1. UV-vis spectra data is very messy. Please suggest additional synthetic conditions for pH as well as AgNO3 solutions and temperature.

2. In order to analyze the FT-IR characteristics of AgNP synthesis results, please present images of two groups of Pseudanabaena/Limnothrix sp cells and AgNP synthesis results.

3. In the case of the antibacterial activity test against E. coli and C. glutamicum, the Disk diffusion test to determined the MIC can derive more accurate results.

Author Response

Referee: 4

  1. UV-vis spectra data is very messy. Please suggest additional synthetic conditions for pH as well as AgNO3 solutions and temperature.

We strongly believe that the UV-vis absorption peaks are clear and give all the appropriate results.

  1. In order to analyze the FT-IR characteristics of AgNP synthesis results, please present images of two groups of Pseudanabaena/Limnothrix sp cells and AgNP synthesis results.

We thank the reviewer for this comment. We added the FT-IR spectra of Pseudanabaena/Limnothrix sp cells. The comparison of the FT-IR spectra shows the relative shifts in the peak positions and intensity distribution, indicating the role of cyanobacterium extract to AgNPs formation. Figure 5 has been modified and placed in the draft. The following sentence has been added as well in the draft.

“The comparison of the FT-IR spectra shows the relative shifts in the peak positions and intensity distribution, indicating the role of cyanobacterium exctract to AgNPs formation.” (Line 258-262)

  1. In the case of the antibacterial activity test against E. coli and C. glutamicum, the Disk diffusion test to determined the MIC can derive more accurate results.

The method of disk diffusion tests constitutes a classic and wide-used method for the calculation of MIC. In the current study, we wanted to test many different concentrations of the produced AgNPs. Nanoparticles have a specific size and as a result we wanted to have a fulltime interaction between nanoparticles and bacterial test during our experiment. This could be achieved with continues shaking of the micro-plates where antibacterial tests took place, while it couldn’t be achieved with disk diffusion test. Moreover, especially for the low nanoparticle concentrations, the disk diffusion test couldn’t provide as accurate results as the dilution method. Finally, as the antibacterial tests took place more than three times as to indicate accurate results, with dilusion method we used quite less amount of AgNps, compared with the needs for disk diffusion tests.

Reviewer 5 Report

Journal: Nanomaterials

Manuscript ID: nanomaterials-1746126

Title: Green synthesis of silver nanoparticles with high antibacterial

activity using cell extracts of cyanobacterium Pseudanabaena/Limnothrix sp.

Authors: Dimitra Karageorgou, Panagiota Zygouri, Theofylaktos Tsakiridis,

Mohamed Amen Hammami, Nikolaos Chalmpes, Mohammed Subrati, Ioannis Sainis,

Konstantinos Spyrou *, Petros Katapodis *, Dimitrios Gournis *, Haralambos

Stamatis

 In this manuscript submitted as Article to Nanomaterials journal, the authors reported synthesis of silver nanoparticles (using as reducing agent a cellular extract of cyanobacterium Pseudanabaena/Limnothrix sp.), their extensive characterization and their antibacterial effects on two bacterial strains (E. coli and C. glutamicum). The topic of this manuscript is relevant to the field of the Nanomaterials and fits with the scope of this journal. The study was well designed and conducted. The text is in general well written (apart from some ambiguous formulations, and grammar or typing errors), in an appropriate style (but in a hurry) and is very concise. The Abstract is enough detailed and Introduction section presents relevant information from the literature (but must be improved). The presentation of methodology is appropriate (but some aspects have to be clarified), and the results are very clear presented using illustrative figures (that support some improvements). The conclusions are based on the obtained results. The authors used a reasonable number of references for preparing the manuscript, most of them recently published, indicating the actuality of the subject. However, before recommending the publication into the Nanomaterials journal, a major revision of the manuscript is required. One major and twenty-two minor aspects remain to be solved.

Major point

According to Instructions for Authors (Manuscript Preparation), the manuscript has to include also a Discussion section among the Research manuscript sections. Therefore, the authors have to add this section to their manuscript, and to move there all small discussions from the current Results section (lines 173-176, 183-190, 194-197, 201-203, 225-226, 230-231, 234-236, 307-309). The authors have to complete the discussion, by comparing the results obtained in this study by various methods, and by comparing their results with those previously published into the literature. In this section, the authors have to (try to) explain the mechanisms responsible for the bactericidal results of their AgNPs, by establishing correlations between the chemical and biological data. They also have to highlight the relevance of their findings (taking into account that AgNPs were also produced with other cyanobacteria, and tested against bacteria), as well as the originality of their study (maybe the last paragraph in Introduction could be moved here). In Results should remain only the results obtained by the authors.

Minor points

1. The authors have to change the title into “Green synthesis and characterization of silver nanoparticles with high antibacterial activity using cell extracts of cyanobacterium Pseudanabaena/Limnothrix sp.” since the large majority of the manuscript consists of characterization of the AgNPs.

2. In Introduction, the authors have to explain in more detail why they decided to test their AgNPs on the two selected bacterial strains (as they well explained the reasons for using Pseudanabaena).

3. The authors have to define all acronyms the first time they appear in each of three sections: the abstract; the main text; the first figure or table (lines 74, 91, 210).

4. In Materials and methods, the authors have to provide the full origin of the reagents, and apparatuses used (company, country, [state], and city) in lines: 90, 102, 103, 105, 121, 130, 132, 134, 138, 147, 155.

5. In Materials and methods and in Results, the authors have to use only Past Tense (lines: 157, 163, 164, 168-171, 193, 228, 236, 238-240, 242-245, 247, 253-258, 285-287, 297-310, 318-325, 333). And the same for Abstract

6. Please rephrase the text in lines: 34, 43,76, 79-82, 185, 187-188, 197, 199 (no PE and PC were actually presented in Fig. 2), 213-214 and 216 [gbiomass], 292-293, 295, 301.

7. In Materials and methods, the authors have to provide the rotor radius of the devices used for shaking or centrifugation, or to express the conditions of centrifugation in g instead of rpm (lines: 94, 96, 104, 110, 119, 120, 124, 153.

8. Please remove “pH” in line 124.

9. The authors should transfer lines 161-163 to Methods.

10. It is not clear from Fig. 1 how temperature influenced the synthesis of AgNPs as stated in line 164.

11. In line 173, is cited ref. Tomer et al. without any number, and without listing it into the References list.

12. In line 226 the correct order is [20][21].

13. In Fig. 7 the authors have to label the panel with a, b, c and d, and to group the four panels into a single figure (by changing also the position of Table 2).

14. The title of table 2 is not illustrative. What surface was analyzed? Of the AgNPs? And why that surface contains much more C than Ag?

15. Please remove “in detail” in line 284 since the AFM images are much better than those obtained by TEM.

16. What is the reason of labelling the four panels in figure 9 as a-d? there are differences between the four? Also the scale bars in Fig. 9 are not visible.

17. After Figure 9, the authors present and discuss Figure 11. They have to change the numbers of the figures following Fig. 9, or to introduce a figure 10.

18. In line 304, it is not clear to what “the same effect” referred the authors. Please be more specific.

19. The authors have to introduce labels (a, b) in the panels of figure 11.

20. It is not clear how the authors calculated LC50 of AgNPs at values higher than 50 µg/mL, when they tested the AgNPs at maximal concentrations of 50 µg/mL. Please explain.

21. There is no consistency in formatting of 9 references from the list (1, 3, 8, 16, 19, 22, 29, 31, 33).

22. In line 394 please writhe Phormidium with capital P.

Author Response

Referee: 5

  1. According to Instructions for Authors (Manuscript Preparation), the manuscript has to include also a Discussion section among the Research manuscript sections. Therefore, the authors have to add this section to their manuscript, and to move there all small discussions from the current Results section (lines 173-176, 183-190, 194-197, 201-203, 225-226, 230-231, 234-236, 307-309). The authors have to complete the discussion, by comparing the results obtained in this study by various methods, and by comparing their results with those previously published into the literature. In this section, the authors have to (try to) explain the mechanisms responsible for the bactericidal results of their AgNPs, by establishing correlations between the chemical and biological data. They also have to highlight the relevance of their findings (taking into account that AgNPs were also produced with other cyanobacteria, and tested against bacteria), as well as the originality of their study (maybe the last paragraph in Introduction could be moved here). In Results should remain only the results obtained by the authors.

We thank the reviewer for this comment. However, the instruction for authors file encourage the authors to present a discussion part that may be combined with the Results, as we did. We strongly believe that in the current work, these two parts should be unseparated as to be easily read and understandable by the readers who will show their interest. A separate discussion part could cause chaos. For these reasons, and according with the advice of the reviewer we enriched the result and discussion part by comparing our results with previous literatures. Moreover, we enrich  the Conclusion part where we highlighted the originality of our work.

The added parts are presented in the draft

“Results and Discussion” (Line 171)

“On the other hand our results come in contrast with previous reported studies where alkali pH values were seemed as more appropriate for the synthetic process.” (Line 185-187)

“This concentration constitutes the lowest silver nitrate amount compared to other works” (Line 193-194)

"the use of the lowest concentration of 0.5 mM AgNO3 ,  20oC and pH 7 in subsequent experiments, minimize the wastes and reduce the production cost, making the whole synthetic process more eco-friendly.” (Line 191-193)

“cells and the synthesized AgNPs” Line 266

“The comparison of the FT-IR spectra show the relative shifts in the peak positions and intensity distribution, indicating the role of cyanobacterium extract to AgNPs formation.” (Line 258-260)

Our results agreed with those of El-Naggar et al and Hamouda et al, who revealed that cell extracts molecules may contribute to the reduction of Ag+(Line 260-263)

“The spherical size of our produced AgNPs comes in accordance with previous studies, where spherical AgNPs are produced” (Line 299-301)

“On contrast, out AgNPs are small in size and appear homogeneity, something that is absent from previous reported AgNPs. For instance, the AgNPs synthesized from Microcoleus sp. Cells were 44-79nm, from Synechococcus sp.
had a size of 15.2-266.7nm, from Anabaena doliolum were 10-50nm, from Leptolyngbya sp were 20-35nm” (Line 301-305)

“The produced biogenic AgNPs are environmentally safer compared to them produced with classic chemical and physical methods, as the synthetic conditions are gendler, with no energy consumption- the synthesis achieved in room temperature of 20oC and there is absent of toxic compounds. Moreover, through the use of Pseudanabaena/Limnothrix sp cells, a new perspective for the overburdened eutrophic cyanobacterium habitant is offered.” (Line 367-372)

  1. The authors have to change the title into “Green synthesis and characterization of silver nanoparticles with high antibacterial activity using cell extracts of cyanobacterium Pseudanabaena/Limnothrix sp.” since the large majority of the manuscript consists of characterization of the AgNPs.

We agree with the Reviewer and the Title has been modified according to his suggestion since a big part of the draft refers to synthesis and characterization.

  1. In Introduction, the authors have to explain in more detail why they decided to test their AgNPs on the two selected bacterial strains (as they well explained the reasons for using Pseudanabaena).

The rationale for using these the bacterial strains has been already reported in Introduction in page 2 and lines 80-82. Briefly, we wanted to test the bactericidal action of the produced AgNPs on gram+ and gram- bacterial cells, because of the differenced on their membranes. E. coli constitutes a characteristic gram- pathogenic bacterium, while C. glutamicum is a characteristic gram+ bacterium widely used in industry. As we propose the produced AgNPs as a possible candidate for the production of antimicrobial tools, we did a first bactericidal test.

  1. The authors have to define all acronyms the first time they appear in each of three sections: the abstract; the main text; the first figure or table (lines 74, 91, 210).

Acronyms defined in the abstract

  1. In Materials and methods, the authors have to provide the full origin of the reagents, and apparatuses used (company, country, [state], and city) in lines: 90, 102, 103, 105, 121, 130, 132, 134, 138, 147, 155.

We could supply these information if needed but for this we need few days to search all the details of the reagents.

  1. In Materials and methods and in Results, the authors have to use only Past Tense (lines: 157, 163, 164, 168-171, 193, 228, 236, 238-240, 242-245, 247, 253-258, 285-287, 297-310, 318-325, 333). And the same for Abstract

We agree with the reviewer, and we change the draft using Past Tense. All small changed are marked in red font.

  1. Please rephrase the text in lines: 34, 43,76, 79-82, 185, 187-188, 197, 199 (no PE and PC were actually presented in Fig. 2), 213-214 and 216 [gbiomass], 292-293, 295, 301.

We rephrase the text lines in most of the lines suggested from the reviewer. All changes are mentioned in red font.

  1. In Materials and methods, the authors have to provide the rotor radius of the devices used for shaking or centrifugation, or to express the conditions of centrifugation in g instead of rpm (lines: 94, 96, 104, 110, 119, 120, 124, 153.

We added the proposed information accordingly to the reviewer comment.

Line 90-91: “MRC Refrigerated Shaker Laboratory Incubator, 600 X 480 mm plate, 0-60 deg c, 250rpm”

Line 102: “MRC Laboratory Shaker Incubator, 450 X 450 mm, 400rpm, 70 deg c”

Line 108: “with 16,4 cm rotor radius”

Line 117: “with 16,4 cm rotor radius”

Line 126-127: “with 8,9cm the rotor radius of the inner ring and 10,1cm of the outer ring”

Line 161: “MRC Laboratory Shaker Incubator, 450 X 450 mm, 400rpm, 70 deg c”

  1. Please remove “pH” in line 124.

pH has been removed in line 124

  1. The authors should transfer lines 161-163 to Methods.

Lines 161-163 have been transferred to Methods

  1. It is not clear from Fig. 1 how temperature influenced the synthesis of AgNPs as stated in line 164.

We checked the line 164 and we totally agree with the comment of the reviewer. In figure 1, we present the effect of pH. The temperature has been removed from the text.

“the synthesis is influenced by pH of the reaction medium” (Line 176)

  1. In line 173, is cited ref. Tomer et al. without any number, and without listing it into the References list.

The reference has been added to the manuscript both on line 183 and in the reference list

“Cyanobacterial extract-mediated synthesis of silver nanoparticles and their application in ammonia sensing” (Line 439)

  1. In line 226 the correct order is [20][21].

Corrected

  1. In Fig. 7 the authors have to label the panel with a, b, c and d, and to group the four panels into a single figure (by changing also the position of Table 2).

We agree with the reviewer. We label the panel with a, b, c and d and we change position to the table.   

  1. The title of table 2 is not illustrative. What surface was analyzed? Of the AgNPs? And why that surface contains much more C than Ag?

The surface that was analyzed is the one of the Ag Nanoparticles. As we demonstrated from AFM there are no impurities, so what we observe in the XPS is the AgNPs. XPS is a surface technique that can take information for the first nm of a surface. What the X-rays see first is the functional carbon that encloses the spherical surface. This is the reason carbon has high intensity. However we have to take into account that in XPS when we want to do atomic percentage estimation we have to calculate as well the sensitivity factor of each element, so the survey is more like a scan to see what elements compose the surface. For more information (atomic bonds, oxidation states, atomic percentages etc.) we are doing high resolution analysis in each element.

  1. Please remove “in detail” in line 284 since the AFM images are much better than those obtained by TEM.

The phrase “in detail” has been removed

  1. What is the reason of labelling the four panels in figure 9 as a-d? there are differences between the four? Also the scale bars in Fig. 9 are not visible.

We would like to thank the reviewer for this question. Indeed there are not many differences in images from a to d. It is different areas we took from TEM in order to show that our final synthesized AgNPs are spherical, homogeneous and have the same size in different areas investigated. This is something that the AFM also confirms. We could remove the three of them by Editor choice. For this reason we add in in Figure 9 line 311 the following:

“The four regions represents four different measurements indicating”

  1. After Figure 9, the authors present and discuss Figure 11. They have to change the numbers of the figures following Fig. 9, or to introduce a figure 10.

Numbers have changed. We apologize for this.

  1. In line 304, it is not clear to what “the same effect” referred the authors. Please be more specific.

In 3.4 part of our results, we discuss the bactericidal effect of AgNPs on two different bacterial strains (E.coli and C. glutamicum). This effect is depended on both the dose of AgNPs and the interaction time of AgNPs-cells. In line 329, it is referred that 30% lethal effect is presented at 10μg/ml AgNPs after 12h of interaction on E.coli cells. The same effect (70% lethal effect) was observed on C.glutamicum cells at 20μg/ml after 2 hours of interaction. Moreover the same effect (70% lethal effect) was observed on C.glutamicum cells at 50μg/ml after 1 hours of interaction.

In line 329, we added the explanation of “70% lethal effect” according to the reviewer comment.

  1. The authors have to introduce labels (a, b) in the panels of figure 11.

Figure 11 has been modified. The figures are placed in different order and not parallel and are distinguished in the text with the terms above and below. Figure 10 now looks clearer and more understandable.

  1. It is not clear how the authors calculated LC50 of AgNPs at values higher than 50 µg/mL, when they tested the AgNPs at maximal concentrations of 50 µg/mL. Please explain.

Median lethal concentrations (LC50) of AgNPs against E. coli and C. glutamicum were evaluated for each exposure time using the calculator freely available at   https://www.aatbio.com/tools/lc50-calculator (AAT Bioquest, Inc. (2022, June 5). Quest Graph™ LC50 Calculator. AAT Bioquest.  https://www.aatbio.com/tools/lc50-calculator)

  1. There is no consistency in formatting of 9 references from the list (1, 3, 8, 16, 19, 22, 29, 31, 33).

References were formatted

  1. In line 394 please writhe Phormidium with capital P.

It is done.

Round 2

Reviewer 2 Report

The authors did not improve the Discussion Section. It is still unknown why the silver nanoparticles produced by the developed method are better than the others.

Please improve the graphs. Dots should be used  on axes – please pay attention to Figure 1-3

Author Response

  1. The authors did not improve the Discussion Section. It is still unknown why the silver nanoparticles produced by the developed method are better than the others.

We thank the reviewer for this comment. However, we would like to mention that the discussion part has been improved in the last version, as we have added literature, that prove the novelty of our work compared to previous ones.

Lines 185-186 indicate the gendle conditions of our synthetic prosess, compared to previous studies.

Lines 190-195 prove the use of extra low AgNO3 concentration.

Linew 300-306 indicate the formation of extra small AgNPs with great homogeneity, something that isn’t referred before.

Last but not least, we have also added the above novel results in our conclusion part.

In summary, the first part of our novelty is the use of these cells (Pseudanabaena/Limnothrix sp) that yield a dual environmental benefit; the alleviation of an overburdened ecosystem and the production of high-value product. Continuing, we would like to mention that we used a green ecofriendly synthetic process, where just 0.5mM of AgNO3, neutral pH and temperature of 20oC were selected as the optimal for our synthesis, compared with the above mentioned papers where extremely alkali pH values, high temperatures (40oC) and AgNO3 concentrations of 10-20mM (which means 20-40 times greater than ours) were used. So, we used milder, less costly and more eco-friendly conditions. Moreover, we produced spherical AgNPs 7-13nm, compared to the produced 7-26nm nanoparticles of the reported work, revealing greater homogeneity.

  1. Please improve the graphs. Dots should be used on axes – please pay attention to Figure 1-3

Figures 1-3 have been modified with dots according to reviewer suggestion.

Reviewer 3 Report

The author made a great effort to answer our questions one by one, and we suggest the journal to accept the manuscript. It is recommended that the English in the manuscript need to be modified. 

Author Response

  1. The author made a great effort to answer our questions one by one, and we suggest the journal to accept the manuscript. It is recommended that the English in the manuscript need to be modified. 

We thank the reviewer for this comment. The manuscript has been checked line by line and modified to improve the English. All changes are with red font.

Reviewer 4 Report

The authors modified the reviewer's request appropriately.

Author Response

We would like to thank the reviewer for accepting the revised version

Reviewer 5 Report

Dear authors,

The revised form of the manuscript is much improved. However, the minor points 3,4,5,6 and 21 were not solved. In these circumstances, I would recommend its publication into Nanomaterials if the Editor considers they could be solved during the editing process.

Author Response

  1. In Introduction, the authors have to explain in more detail why they decided to test their AgNPs on the two selected bacterial strains (as they well explained the reasons for using Pseudanabaena).

One of the possible applications for the produced AgNPs is the development of cover surfaces (eg workbenches, knobs, etc.) with strong antibacterial properties can serve to limit the spread of bacteria that dominate the workplace, such as in hospitals and industrial units. So, we chose a pathogenic bacterial strain, that is met in hospitals and another one that is met in industry. We should mention that it’s an ongoing project and we are going to test much more bacterial strains. These two were our first option for the above reasons.

  1. The authors have to define all acronyms the first time they appear in each of three sections: the abstract; the main text; the first figure or table (lines 74, 91, 210).

All acronyms have been defined in each of three sections. See manuscript

  1. In Materials and methods, the authors have to provide the full origin of the reagents, and apparatuses used (company, country, [state], and city) in lines: 90, 102, 103, 105, 121, 130, 132, 134, 138, 147, 155.

The full origin of the reagents has been added in the manuscript.

BG-11 medium (Fluka Analytical - Switzerland

Luria Bertani medium (LAB173) - Lancashire, UK

LB agar (LAB, a Neogen company) – Lancashire, UK

AgNO3: Sigma-Aldrich St. Louis, MO, USA

JASCO, V-730 spectrophotometer – Tokyo, Japan

FTIR-8400 spectrometer (Shimadzu) –Kyoto, Japan

  1. In Materials and methods and in Results, the authors have to use only Past Tense (lines: 157, 163, 164, 168-171, 193, 228, 236, 238-240, 242-245, 247, 253-258, 285-287, 297-310, 318-325, 333). And the same for Abstract

For all lines referred in the manuscript from the reviewer we used the Pasr Tense

  1. DghfdIt is not clear how the authors calculated LC50 of AgNPs at values higher than 50 µg/mL, when they tested the AgNPs at maximal concentrations of 50 µg/mL. Please explain.

We used a computational method to derive the median lethal concentration (LC50) from concentration-mortality data and not a graphical one. So, it is not necessary for the tested AgNPs concentrations to be higher than the calculated LC50 in every case. The LC50 and its 95 percent confidence limits can be calculated from the data produced by the mortality tests